# MC-DARTS : Model Size Constrained Differentiable Architecture Search

**Kazuki Hemmi**
University of Tsukuba /
National Institute of Advanced Industrial
Science and Technology (AIST)
henmi-kazuki@aist.go.jp

**Yuki Tanigaki**
National Institute of Advanced Industrial
Science and Technology (AIST)
tanigaki.yuki@aist.go.jp

**Masaki Onishi**
National Institute of Advanced Industrial
Science and Technology (AIST)
onishi-masaki@aist.go.jp

## Abstract

Recently, extensive research has been conducted on automated machine learning(AutoML). Neural architecture search (NAS) in AutoML is a crucial method for automatically optimizing neural network architectures according to applying data and its usage. One of the prospected ways to search for a high accuracy model is the gradient method NAS, known as differentiable architecture search (DARTS). Previous DARTS-based studies have proposed that the size of the optimal architecture depends on the size of the dataset. If the optimal size of the architecture is small, the search for a large model size architecture is unnecessary. The size of the architectures must be considered when deep learning is used on mobile devices and embedded systems since the memory on these platforms is limited. Therefore, in this paper, we propose a novel approach, known as model size constrained DARTS. The proposed approach adds constraints to DARTS to search for a network architecture, considering the accuracy and the model size. As a result, the proposed method can efficiently search for network architectures with short training time and high accuracy under constrained conditions.

## 1 Introduction

Recently, deep learning has been widely used in many fields, such as image recognition [1, 2], speech recognition [3], and natural language processing [4], due to its high recognition accuracy. Neural network architectures in deep learning are becoming vast and more complex. Neural network architectures originally inspired by the human brain are used in deep learning to learn the features and rules of a given task. Adjustments to the network architectures and parameters have a significant impact on the accuracy of the task. However, manual adjustments of neural network architectures are exceedingly difficult due to their complexity. As the evaluation cost of deep learning is more expensive, the number of possible trial and error times will be less. For these reasons, automated machine learning (AutoML), which automatically adjusts network parameters and architectures, has been actively studied [5]. This study focuses on neural architecture search (NAS) in AutoML. NAS is a method that automatically searches for the optimal network architecture.

In a previous study [6], experiments using the NAS method, MiLeNAS, confirmed that increasing the model size beyond the optimal size does not improve the performance. Architecture search for a larger than optimal model size is unnecessary because optimal network architectures cannot be searched. Hence, in this paper, we considered that we can efficiently search for the optimal architecture by constraining the search to skip too large model size architectures. Additionally, in an environment with limited memory capacity, a network architecture needs to be lightweight.

Has it Trained Yet? Workshop at the Conference on Neural Information Processing Systems (NeurIPS 2022).

In other words, the model size of network architectures is a crucial factor in practical application. Therefore, we propose model size constrained differentiable architecture search (MC-DARTS), a novel approach based on the DARTS [7] of the gradient method NAS. The proposed approach can search for a network structure that fits the objective while optimizing with constrained model size.

The contributions of this study can be summarized as follows: (1) We propose a NAS method that can search the network architecture using a constrained model size. (2) We perform image classification experiments using the proposed method. The results of the experiments show that the proposed method can search for small network architectures compared with the usual unconstrained NAS method, and the use of the proposed method results in reduced training time and improved classification accuracy. (3) Finally, we analyze the type of constraints that are most efficient by setting multiple constraint conditions.

To encourage open science, we share our codes and the network architectures which have been searched with the public: https://github.com/itigo-11111/MC-DARTS

## 2 Related Work

### 2.1 NAS

Recently, there has been active research on how NAS can automatically search for the optimal network architecture for a specific dataset. NAS was first proposed in 2016. At first, the mainstream of the NAS method was reinforcement learning [8, 9, 10, 11, 12] and optimization of the evolutionary algorithm [13, 14, 15, 16]. However, these methods are very computationally expensive, requiring more than 2000 GPU days to search for architectures. Zoph & Le (2018) proposed a Cell-based NAS-Net [17] that stacks small blocks of a directed acyclic graph instead of the entire network architecture, to reduce computational cost in 2018. Nodes in a directed acyclic graph in NAS-Net correspond to feature maps, while its edges correspond to candidate operations.

### 2.2 Differentiable Architecture Search (DARTS)

DARTS [7] is the NAS-Net-based method. DARTS enables effective architecture search using weight sharing and gradient methods to convert to continuous search space. Weight sharing is the technique used in One-Shot NAS to simultaneously optimize architectures and train the networks. When selecting candidate architectures from a supernet containing all searchable network architectures, it is possible to reduce the computational cost required for the architecture selection by reusing the parameters that the supernet is learning. DARTS optimizes two kinds of cells (normal and reduction cells). Node $\boldsymbol{x}^{(j)}$ in each cell is defined as $\boldsymbol{x}^{(j)} = \sum_{i<j} \overline{o}^{(i,j)}(\boldsymbol{x}^{(i)})$. During the search, the existence of an edge between two nodes is determined using the equation given below:

$$\overline{o}^{(i,j)}(\boldsymbol{x}^{(i)}) = \sum_{o \in \mathbf{O}} \frac{\exp(\alpha_o^{(i,j)})}{\sum_{o' \in \mathbf{O}} \exp(\alpha_{o'}^{(i,j)})} o(\boldsymbol{x}^{(i)}) \tag{1}$$

where $\mathbf{O}$ is a candidate operation (e.g., Skip Connection) entering an edge, and the architecture weight of a pair of nodes $(i, j)$ is a vector $\boldsymbol{\alpha}^{(i,j)}$ of dimension $|\mathbf{O}|$. The architecture weights $\boldsymbol{\alpha}$ a represent the significance of each candidate operation in determining the network architecture. When the search is completed, the architecture is determined using $o^{(i,j)} = \underset{o \in \mathbf{O}}{\operatorname{argmax}} \, \boldsymbol{\alpha}_o^{(i,j)}$. DARTS searches for network architectures using architecture weights and training parameters optimized to minimize loss. Next, DARTS optimizes the training parameters of the search architectures to minimize loss. DARTS can achieve higher accuracy than previous methods because it retrains after the architecture search. After DARTS was proposed, some comparable methods have been proposed [18, 19, 20, 21, 22, 23]. DARTS became a baseline for NAS. There are DARTS-based methods [24, 25, 26] that are constrained to computation time, but very few DARTS-based methods are constrained to model size.

## 3 Proposed Approach

In this study, we propose a constrained optimization method using two kinds of techniques. **Alg. 1** shows an overview of the algorithm for the constrained optimization method.

---

**Algorithm 1** MC-DARTS : Model Size Constrained Differentiable Architecture Search

---

1: Create a mixed operation $\overline{o}^{(i,j)}$ parametrized by $\boldsymbol{\alpha}^{(i,j)}$ for each edge $(i, j)$
2: **while** the current number of learning times has not achieved the set epoch, **do**
3:     **if** no change either in architecture from the previous iteration or not achieved constraint condition, **then**
4:         Adjust the priority in $\boldsymbol{\alpha}$ ( 3.2 Change Alpha's Priority )
5:     **if** constraint condition is not achieved, **then**
6:         $\boldsymbol{\beta}$ = convolutional weights in $\boldsymbol{\alpha}$
7:         Update architecture weights $\boldsymbol{\beta}$ by descending $\nabla_{\boldsymbol{\beta}} \mathcal{L}_{val}^{MC}(w^*(\boldsymbol{\alpha}), \boldsymbol{\beta})$
8:         Update $\boldsymbol{\alpha}$ by Concat($\boldsymbol{\beta}$, $\boldsymbol{\gamma}$ = non-convolutional weights in $\boldsymbol{\alpha}$)
9:     **else**
10:         Update architecture weights $\boldsymbol{\alpha}$ by descending $\nabla_{\boldsymbol{\alpha}} \mathcal{L}_{val}(w^*(\boldsymbol{\alpha}), \boldsymbol{\alpha})$     (Eq. (2) vanilla DARTS)
11:     Update training parameter $\boldsymbol{w}$ by descending $\nabla_{\boldsymbol{w}} \mathcal{L}_{train}(\boldsymbol{w}, \boldsymbol{\alpha})$
12: Derive the final architecture based on the learned $\boldsymbol{\alpha}$.

---

## 3.1 MC-DARTS : Model Size Constrained Differentiable Architecture Search

MC-DARTS is an improved DARTS-based method. MC-DARTS searches for optimum network architectures under given constraints by setting model size constraints beforehand. MC-DARTS replaces the loss function using the model size constraint as follows:

$$\min_{\boldsymbol{\alpha}} \mathcal{L}_{val}(w^*(\boldsymbol{\alpha}), \boldsymbol{\alpha}) \Longrightarrow \begin{cases} \min_{\boldsymbol{\beta} \in \boldsymbol{\alpha}} \mathcal{L}_{val}^{MC}(w^*(\boldsymbol{\alpha}), \boldsymbol{\beta}), & \text{if } \mathcal{M}(w^*(\boldsymbol{\alpha})) \geq \mathcal{M}_c \\ \min_{\boldsymbol{\alpha}} \mathcal{L}_{val}(w^*(\boldsymbol{\alpha}), \boldsymbol{\alpha}), & \text{if } \mathcal{M}(w^*(\boldsymbol{\alpha})) < \mathcal{M}_c \end{cases}$$

$$\text{where } \min_{\boldsymbol{\beta} \in \boldsymbol{\alpha}} \mathcal{L}_{val}^{MC} = \lambda(\mathcal{M}(w^*(\boldsymbol{\alpha})) - \mathcal{M}_c)\mathcal{L}_{val}(w^*(\boldsymbol{\alpha}), \boldsymbol{\beta}) \tag{2}$$

$$\text{s.t. } w^*(\boldsymbol{\alpha}) = \underset{\boldsymbol{w}}{\operatorname{argmin}} \ \mathcal{L}_{train}(w(\boldsymbol{\alpha}), \boldsymbol{\alpha})$$

where $\mathcal{M}$ is the current model size, $\mathcal{M}_c$ is the constraint condition, $\boldsymbol{w}$ are the network parameters, $\boldsymbol{\beta}$ are the architecture weights for the convolutional layers, $\boldsymbol{\gamma}$ are the other architecture weights, $\mathcal{L}_{train}$ is the loss for training data, $\mathcal{L}_{val}$ is the loss for validation data, and $\lambda$ is hyperparameter to control the speed at which the $\mathcal{L}_{val}^{MC}$ weights are updated. Until the constraint is achieved, MC-DARTS only optimizes the architectural weights $\boldsymbol{\beta}$ for the convolutional layers affecting model size. $\mathcal{L}_{val}^{MC}$ is used only when the model size does not achieve the constraint. Hence, updating architecture weights using $\mathcal{L}_{val}^{MC}$ always results in $\mathcal{M}(w^*(\boldsymbol{\alpha})) \geq \mathcal{M}_c$. If the constraint condition is achieved, MC-DARTS is updated as a conventional DARTS.

MC-DARTS calculates the gradient of the architecture using the following Eq. (3). Note that in this study, the computational part of the second-order differentiation is assumed to be $\xi = 0$ because of the learning time needed.

$$\nabla_{\boldsymbol{\alpha}} \mathcal{L}_{val}(w^*(\boldsymbol{\alpha}), \boldsymbol{\alpha}) \approx \nabla_{\boldsymbol{\alpha}} \mathcal{L}_{val}(\mathbf{w} - \xi \nabla_{\boldsymbol{w}} \mathcal{L}_{train}(\mathbf{w}, \boldsymbol{\alpha}), \boldsymbol{\alpha}) \tag{3}$$

## 3.2 Change Alpha's Priority

If there is no change either in architecture from the previous iteration or not been achieved constraint condition, Change Alpha's Priority performs the following operations:

(1) Randomly select node $(i, j)$ and extract the architecture weights $\boldsymbol{\alpha}^{(i,j)}$ of all candidate operations between the nodes.

(2) Check whether the operation corresponding to the largest architecture weight in $\boldsymbol{\alpha}^{(i,j)}$ has parameter $\boldsymbol{w}$, and return to (1) if it does not (limited number of returns).

(3) Set the architecture weight of $\alpha^{'(i,j)}$ selected in (2) to the same value as the $N$th largest architecture weight between the same nodes ($N$ depends on the number of convolution operations of the operation candidates, $N = 5$ in the case of DARTS).

(4) Perform operations (1)–(3) for both normal and reduction cells.

During the optimization, Change Alpha's Priority increases the frequency of architectural changes. Therefore, MC-DARTS can search for architectures in a more extensive search space.

Table 1: Experimental result for each architectures, with the best results in bold + underline and the next best results in bold

| Architecture | Constraint(MC) (M) | Model Size (M) | Best Val Acc. (%) | Mean Latency (ms) | Total Time (h) |
|---|---|---|---|---|---|
| ResNet-50 [27] | - | 23.52 | 89.62 | 14.63 | - |
| ShuffleNet-v2-x2.0 [28] | - | 5.36 | 85.43 | 14.88 | - |
| (DARTS) | - | 3.05 | **95.17** | 47.54 | 22.90 |
| | 4.00 | 2.78 | 94.50 | 44.05 | 22.06 |
| | 2.90 | 2.67 | **95.63** | 40.06 | 21.26 |
| | 2.60 | 2.57 | 94.97 | 38.20 | 20.57 |
| MC-DARTS | 2.30 | 2.07 | 94.59 | 30.86 | 17.26 |
| | 2.00 | 1.83 | 93.67 | 28.83 | 15.93 |
| | 1.70 | **1.63** | 93.97 | **21.84** | **14.07** |
| | 1.50 | **1.39** | 88.11 | **20.20** | **13.13** |

# 4 Experiments

## 4.1 Dataset and Experimental Setting

The experimental dataset used in this study was CIFAR10 with 10 types of classes. In the experiments, we used the same search space as in the original DARTS paper [7], with the operation candidate $|\mathbf{O}| = 8$. The image size of the dataset is $32 \times 32$, the number of cells for the architecture search is 8, and the number of cells for the architecture evaluation is 20. The other hyperparameter settings for MC-DARTS are shown in the Appendix. The experimental setting is the same as that in the original DARTS paper [7]. We applied the model size constraints: 1.5, 1.7, 2.0, 2.3, 2.6, 2.9, and 4.0 M. The two main evaluations of the experiment are (1) whether it is possible to search for architectures with smaller model sizes than the unconstrained DARTS, and (2) whether classification accuracy changes when the model size is reduced. Additionally, we investigated whether constraining the model size with multiple constraints could be more efficient.

## 4.2 Results and Discussion

**Figure 1** in the appendix shows the search process, **table 1** shows the results of the architecture evaluation of the conventional deep learning architecture and the network architecture searched using the proposed method. The constraint and model size indicate the number of parameters in the architecture, and the mean latency indicates the average inference time per attempt. The total time is the sum of architecture search time and the architecture evaluation time calculated using one NVIDIA Tesla V100 SXM2 16GB. In terms of model size, the network architecture searched under a constrained condition is smaller than the network architecture searched under an unconstrained condition. In terms of classification accuracy, the network architecture with a model size of 1.39 M (constraint:1.50 M), which is the smallest model size, has a significantly lower Best Validation Accuracy than the other network architectures. However, the modest-sized network architecture of 2.67 M (constraint: 2.90 M) outperforms the unconstrained network architecture. Therefore, it is clear that a reasonably large model size is required to achieve high discrimination accuracy. However, it is unclear why the value of 2.90 M is acceptable as a constraint condition, and we would like to investigate this question in the future.

# 5 Conclusion

We proposed a DARTS-based method, MC-DARTS, with model size constraints and performed image classification experiments. The experimental results showed that MC-DARTS can freely search for small network architectures and can compress the network architecture to the pre-specified model size while maintaining classification accuracy. In Appendix A, our proposed method is also combined with another DARTS-based approach, and the results also revealed the same trend with constraints of a more detailed range of the model size. However, the search for smaller network architectures is difficult because the proposed method optimizes the contents of the cell, and there is a limit to the range of constraint settings. Therefore, in the future, we will consider a method that allows for a more flexible constraint setting.

## Acknowledgements

This paper is based on results obtained from a project commissioned by the New Energy and Industrial Technology Development Organization (NEDO).

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

## A  Appendix

### A.1  Partial Channel Connections for Memory-Efficient Architecture Search(PC-DARTS)

PC-DARTS [23] is designed to minimize memory consumption while maintaining accuracy. PC-DARTS uses a method to extract and perform calculations on a part of the channels, as shown in Equation (4). The $\mathbf{S}^{(i,j)}$ in Equation (4) is a mask that assigns 1 to the selected channel and 0 to the masked channel.

$$o_{PC}^{(i,j)}(\mathbf{x_i}; \mathbf{S}^{(i,j)}) = \sum_{o \in \mathbf{O}} \frac{\exp(\boldsymbol{\alpha}_o^{(i,j)})}{\sum_{o' \in \mathbf{O}} \exp(\boldsymbol{\alpha}_{o'}^{(i,j)})} \times o(\mathbf{S}^{(i,j)} \times \mathbf{x_i}) + (1 - \mathbf{S}^{(i,j)}) \times \mathbf{x_i} \qquad (4)$$

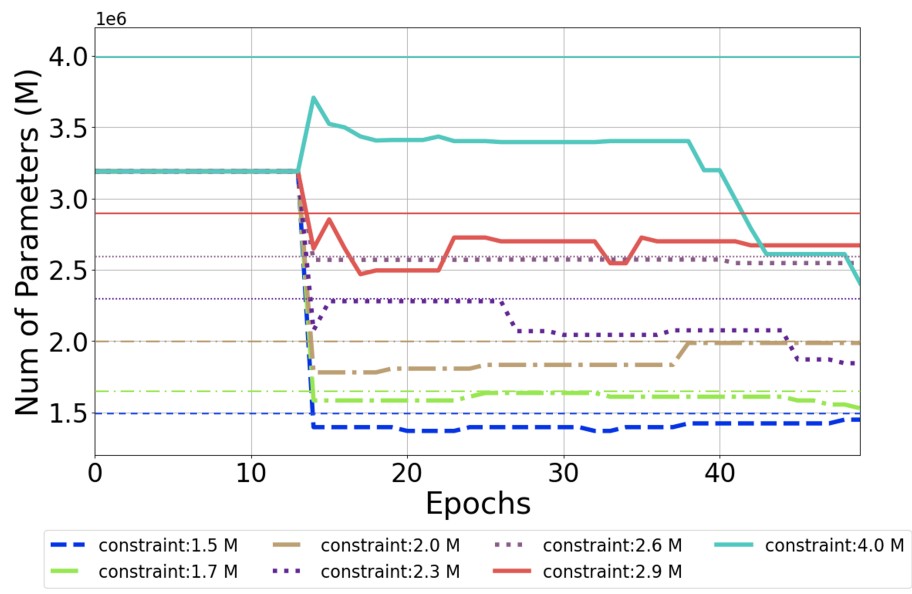

Figure 1: Search process(poly-line) and constraints(straight-line) in MC-DARTS

## A.2 Experiments using MC-PC-DARTS：Model Size Constrained Partial Channel Connections for Memory-Efficient Architecture Search

We examined using MC-PC-DARTS, an extension of PC-DARTS, to confirm whether the non-DARTS method could work. **Table 2** shows the main experimental settings. The experimental setting was the same as that in the original DARTS and PC-DARTS studies[7, 23]. We applied the following model size constraints: 1.4 to 3.0, 3.5, 4.0, 4.5, and 5.0 M. To compare with other existing architectures, we experimented with five different deep learning architectures, including ResNet-50 and ShuffleNet-v2-x0.5, in the same settings as MC-DARTS.

**Table 3** shows the results of the architecture evaluation on the conventional deep learning architecture and the network architecture searched using the proposed method. In terms of model size, the network architecture searched with a constrained condition is smaller than the network architecture searched with an unconstrained condition. In terms of classification accuracy, the modest-sized network architectures of 2.27 M (constraint: 2.80 M), 2.31 M (constraint: 2.50 M), and 1.91 M (constraint: 2.40 M) outperform the unconstrained network architecture. In the above experiments, there were no architectures larger than the model size of 2.90 M, and the search with constraints larger than 2.90 M converged to architectures between 2.19 M and 2.84 M, which is similar in model size to the high-accuracy architectures (constraint: 2.80 M, 3.50 M). However, architectures searched with constraints larger than 2.90 M have lower Best Validation Accuracy than the high-accuracy architectures. In the MC-PC-DARTS, the search space is narrowed by setting modest-small sized constraint conditions. As a result, we assume that MC-PC-DARTS can search a lot around the optimal architecture. In addition, DARTS-based methods simultaneously optimize architectures and train the networks. Consequently, a better validation accuracy is obtained in the second half of the total Epochs. Our proposed method is unavailable for the user to specify the size of the architecture to be trained in the second half of the Epochs. We believe it is necessary to analyze this effect in the future. Eventually, MC-PC-DARTS, an extension of PC-DARTS with the constraint conditions, can search for small network architectures like MC-DARTS. Therefore we confirmed that the network architecture can be compressed.

Table 2: Experimental settings, which conforms to those in DARTS and PC-DARTS papers

| | |
|---|---|
| Image Size of Dataset | $32 \times 32$ |
| Number of Cells(Search) | 8 Cell(6 Normal Cell, 2 Reduction Cell) |
| Number of Cells(Train) | 20 Cell(18 Normal Cell, 2 Reduction Cell) |
| Number of Initial Channels | Search : 16 , Train : 36 |
| Lambda for Architecture | 0.001 |
| Random Seed | 1001 |
| Number of Training Epochs | Search : 50 Train : 600 |
| Loss Function | CrossEntropyLoss |
| Initial Learning Rate(Search) | MC-DARTS：0.025,MC-PC-DARTS：0.1 |
| Initial Learning Rate(Train) | 0.025 |
| Learning Rate Decay | $\rightarrow 0$ (Cosine Scheduler) |
| Learning Rate for Architecture | MC-DARTS：0.0003,MC-PC-DARTS：0.0006 |
| Data Augmentation | Random Crop,Horizontal Flip |
| Cutout | MC-DARTS：False,MC-PC-DARTS：True |
| Optimizer | Training Parameter : SGD,Architecture Weight : Adam |

Table 3: Experimental result for each architecture, with the best results in bold + underline and the next best results in bold

| Architecture | Constraint(MC) (M) | Model Size (M) | Best Val Acc. (%) | Mean Latency (ms) | Total Time (h) |
|---|---|---|---|---|---|
| ResNet-18 | - | 11.18 | 89.22 | 10.42 | - |
| ResNet-50 | - | 23.52 | 89.62 | 14.63 | - |
| ResNet-101 | - | 42.52 | 88.90 | 25.46 | - |
| ShuffleNet-v2-x0.5 | - | 0.35 | 80.23 | 12.87 | - |
| ShuffleNet-v2-x2.0 | - | 5.36 | 85.43 | 14.88 | - |
| (DARTS) | - | 3.05 | **95.17** | 47.54 | 22.90 |
| | 4.00 | 2.78 | 94.50 | 44.05 | 22.06 |
| | 2.90 | 2.67 | **95.63** | 40.06 | 21.26 |
| | 2.60 | 2.57 | 94.97 | 38.20 | 20.57 |
| MC-DARTS | 2.30 | 2.07 | 94.59 | 30.86 | 17.26 |
| | 2.00 | 1.83 | 93.67 | 28.83 | 15.93 |
| | 1.70 | **1.63** | 93.97 | **21.84** | **14.07** |
| | 1.50 | **1.39** | 88.11 | **20.20** | **13.13** |
| (PC-DARTS) | - | 2.53 | 95.21 | 48.62 | 14.68 |
| | 5.00 | 2.20 | 94.84 | 41.77 | 13.19 |
| | 4.50 | 2.19 | 94.32 | 44.14 | 13.08 |
| | 4.00 | 2.21 | 94.47 | 42.43 | 13.31 |
| | 3.50 | 2.77 | 95.21 | 57.54 | 17.46 |
| | 3.00 | 2.42 | 94.72 | 49.10 | 14.81 |
| | 2.90 | 2.84 | 94.71 | 56.46 | 17.76 |
| | 2.80 | 2.27 | **95.26** | 42.24 | 13.25 |
| | 2.70 | 2.44 | 95.13 | 47.54 | 14.97 |
| | 2.60 | 2.56 | 95.01 | 50.18 | 15.93 |
| | 2.50 | 2.31 | 95.22 | 45.06 | 14.52 |
| MC-PC-DARTS | 2.40 | 1.91 | **95.23** | 40.08 | 11.26 |
| | 2.30 | 2.13 | 95.10 | 45.45 | 13.34 |
| | 2.20 | 1.95 | 94.61 | 40.13 | 11.74 |
| | 2.10 | 1.98 | 94.87 | 39.39 | 12.55 |
| | 2.00 | 1.96 | 94.75 | 45.78 | 12.17 |
| | 1.90 | 1.84 | 95.01 | 41.51 | 12.92 |
| | 1.80 | 1.66 | 94.74 | 33.81 | 10.55 |
| | 1.70 | 1.68 | 94.65 | 36.12 | 11.43 |
| | 1.60 | 1.45 | 93.83 | 32.36 | **9.67** |
| | 1.50 | **1.42** | 93.60 | **30.72** | 10.04 |
| | 1.40 | **1.40** | 92.11 | **29.50** | **8.76** |

