# OpenReview forum: "MC-DARTS : Model Size Constrained Differentiable Architecture Search"
_NeurIPS.cc/2022/Workshop/HITY — HITY Workshop NeurIPS 2022_

### Official Review · Reviewer_oaXm · 2022-10-17
**DARTS-based method with constrained model size for neural architecture search in AutoML**

**Rating:** 1
**Confidence:** 2

**Review:**

The authors propose another DARTS-based method for neural architecture search in AutoML with constrained model size. The general presentation of the work is comprehensive, but in many places unclear.
The reviewer recommends to check the work "RC-DARTS: Resource Constrained Differentiable Architecture search" by Jin et al. (2019), which also considers the number of parameters as a constraint. The algorithms in both works seem to be quite different, which is why I propose to accept this work.

---

### Official Review · Reviewer_kLkx · 2022-10-17

**Rating:** 1
**Confidence:** 4

**Review:**

This manuscript proposes a new method for (differentiable) neural architecture search that takes into account model size constraints. The paper is clearly written and the results are compelling, demonstrating gains over previous (e.g. unconstrained) approaches.

---

### Official Review · Reviewer_2yGs · 2022-10-17
**Simple extension to DARTS that allows constraints.**

**Rating:** 1
**Confidence:** 4

**Review:**

The paper proposes an extension to the DARTS method for neural architecture search, by augmenting it with constraints, in particular constraints on the model size and memory consumption. The DARTS method continuously relaxes the combinatorial NAS problem and is hence able to use gradient-based optimizers to find a good architecture. The proposed modification of DARTS simply uses a different loss function that penalizes high model complexity whenever the constraints are not fulfilled. The experiments presented are very limited, even for a workshop paper, and it is hard to draw any conclusion from them. Nevertheless, the problem that the paper aims to address is certainly relevant and the proposed method seems promising enough.

---

### Decision · Program_Chairs · 2022-10-20

Accept